# Minimal exposure durations reveal visual processing priorities for different stimulus attributes

Renzo C. Lanfranco [1,2,3] ✉, Andrés Canales-Johnson [4,5], Hugh Rabagliati [1], Axel Cleeremans[3,7] & David Carmel [1,6,7] ✉

Human vision can detect a single photon, but the minimal exposure required to extract meaning from stimulation remains unknown. This requirement cannot be characterised by stimulus energy, because the system is differentially sensitive to attributes defined by configuration rather than physical amplitude. Determining minimal exposure durations required for processing various stimulus attributes can thus reveal the system's priorities. Using a tachistoscope enabling arbitrarily brief displays, we establish minimal durations for processing human faces, a stimulus category whose perception is associated with several well-characterised behavioural and neural markers. Neural and psychophysical measures show a sequence of distinct minimal exposures for stimulation detection, object-level detection, face-specific processing, and emotion-specific processing. Resolving ongoing debates, face orientation affects minimal exposure but emotional expression does not. Awareness emerges with detection, showing no evidence of subliminal perception. These findings inform theories of visual processing and awareness, elucidating the information to which the visual system is attuned.

Although the human visual system is sensitive to the smallest amounts of light that are physically possible[1,2], it is unclear how much visual input the system requires for the extraction of meaningful information. The more sensitive the visual system is to a visual feature, the less exposure to this feature should be required to process it. Measuring the minimal exposures that are required to perceive different stimulus properties can therefore reveal whether the presence of certain features facilitates processing[3-7] and whether such facilitation is limited to the feature in question or boosts processing of the stimulus as a whole[3,4,8]. Furthermore, minimal exposures can capture the functional and neural distinction between subliminal and supraliminal processing, by comparing the durations required for explicit measures of awareness to those required for objective perceptual judgements and stimulus-evoked neural activity. This can critically inform theories of consciousness, which differ as to whether they propose that awareness arises simultaneously for all aspects of a stimulus. The prominent global neuronal workspace theory (GNWT), for example, postulates the concept of 'ignition'[9,10], whereby once a stimulus reaches the threshold for conscious detection, information about it is shared across the brain. This implies that all stimulus attributes become available to awareness together, predicting a single minimal required exposure for conscious access to all attributes (note that this does not rule out perceptual processing of individual attributes without awareness). In other theoretical formulations (e.g., recurrent[11] and

[1]Department of Psychology, University of Edinburgh, EH8 9JZ Edinburgh, United Kingdom. [2]Department of Neuroscience, Karolinska Institutet, 171 65, Stockholm, Sweden. [3]Consciousness, Cognition & Computation Group, Center for Research in Cognition & Neurosciences, ULB Neuroscience Institute, Université libre de Bruxelles, B1050 Brussels, Belgium. [4]Department of Psychology, University of Cambridge, CB2 2EB Cambridge, United Kingdom. [5]Neuropsychology and Cognitive Neurosciences Research Center, Faculty of Health Sciences, Universidad Católica del Maule, Talca, Chile. [6]School of Psychology, Victoria University of Wellington, 6012 Wellington, New Zealand. [7]These authors jointly supervised this work: Axel Cleeremans, David Carmel. ✉e-mail: Renzo.Lanfranco@ki.se; David.Carmel@vuw.ac.nz

predictive[12] processing, and recent modifications of the GNWT[13]), subjective experience may arise independently for different attributes of the same stimulus; such views can also accommodate the possibility that higher-level processing may occur subliminally, and imply that for any given attribute, the minimal exposures required for detection, identification, and awareness may differ.

Here, we measured minimal required exposures by finding the shortest display duration that evokes behavioural and neural indices of processing, i.e., the minimal bottom-up amount of stimulation necessary to trigger these processes. To date, measurement of such minima has not been possible, as even the briefest presentations offered by current standard displays (~7–16 ms) are not sufficiently brief to prevent detection and identification of complex images. To circumvent this limitation, researchers have employed masking techniques, where visual processing and sometimes awareness are disrupted by co-localised masks. However, masking confounds processing of the masked stimulus with processing of the mask and with stimulus-mask interactions, making it impossible to determine the minimal exposure required for perceptual processing of individual stimuli.

We used a newly-developed LCD tachistoscope that enables fast and highly-precise visual presentations (Supplementary Note 1 and Supplementary Fig. 1)[14]. This enabled us to present stimuli for arbitrarily brief durations, without masking, and measure the durations required to process various aspects of these stimuli. We used a stimulus category that conveys a wealth of information in human social life: human faces. Importantly, face processing is associated with a host of well-characterised behavioural and neural markers whose presence can be used to assess whether various aspects of a face are processed, and whether this processing happened consciously or subliminally.

Previous studies have suggested a processing advantage and faster access to awareness for upright over inverted faces[15–20], and for emotional over neutral faces[21,22] (although the latter claim, in particular, has been challenged[23–26]). If so, processing—and possibly awareness—of upright and emotional faces should require shorter minimal exposures than inverted and neutral faces, respectively.

In this work, we use a combination of behavioural and neural approaches to test these hypotheses. In our first experiments, observers were presented with both intact faces and their scrambled counterparts, equated for luminance and contrast, and we measured the minimal exposure duration that was necessary to detect aspects of meaning in the stimulus. Specifically, we measured sensitivity to the location of the stimulus that contained a face (rather than the noise stimulus containing low-level information consistent with a face), and also measured sensitivity to the emotional identity of that face. The two-alternative forced-choice localisation task meant that observers could only succeed by focusing on the extraction of meaning from stimuli (i.e., parsing the particular arrangement of stimulus features into a face), and the application of signal detection analyses enabled criterion-free measurement of sensitivity to these different stimulus aspects. In addition, our tasks also asked observers to rate their own subjective visual experience on each trial, allowing us to assess the degree to which observers were sensitive to their own perceptions of the stimuli. In subsequent experiments, we augmented these procedures with experiments examining how meaningful stimulus attributes may affect single-stimulus detection (indicating meaning extraction), as well as experiments that collected a set of EEG measures, allowing us to evaluate the match between behavioural sensitivity to stimulus aspects and neural indices of processing. The high-precision bottom-up stimulation provided by the tachistoscope enabled accurate measurement of the minimal exposure durations required for perception of meaning.

## Results
### Psychophysical measures of minimal required exposures
We first investigated the hypotheses psychophysically: In Experiment 1, 32 observers were shown brief displays consisting of an intact face (upright or inverted) on one side of fixation, and its scrambled counterpart on the other side. Intact faces had either a fearful or neutral expression (see Supplementary Note 2 and Supplementary Figs. 2–3). Each display was shown for one of seven equally-spaced durations (range 0.8–6.2 ms). After stimulus offset, participants reported the intact face's location and expression with a single keypress; next, they rated their subjective experience of the display using the four-point perceptual-awareness scale (PAS; Fig. 1A).

Location sensitivity increased with exposure duration, spanning chance-level (d′≈0) to high sensitivity ($F_{(1.99, 61.97)}$ =215.135, $p < 0.001$, $\eta p^2 = 0.874$), (Fig. 1B). Detection of intact faces rose above chance at 1.7 ms for all images (one-sample t-tests against zero, uncorrected; neutral upright: $t(31) = 2.469, p = 0.0019, d = 0.436$, $CI = 0.029 − 0.303$; neutral inverted: $t(31) = 4.174, p < 0.001, d = 0.738$, $CI = 0.137 − 0.398$; fearful inverted: $t(31) = 3.253, p = 0.003, d = 0.575$, $CI = 0.085 − 0.37$), except upright fearful faces, which required 2.6 ms ($t(31) = 7.657, p < 0.001, d = 1.354, CI = 0.46 − 0.794$). An overall advantage for upright over inverted faces ($F_{(1, 31)} = 34.918$, $p < 0.001, \eta p^2 = 0.53$) interacted with exposure duration ($F_{(4.9, 151.98)} = 15.331$, $p < 0.001$, $\eta p^2 = 0.331$), arising at exposures of 4.4 ms ($t(31) = 5.68, p < 0.001, d = 0.671, CI = 0.125 − 0.65$) and above. This upright-face advantage, known as the face-inversion effect (FIE), is considered a marker of configural or holistic processing, where dedicated mechanisms (attuned to upright-canonical face orientations) integrate facial features[15–17]. Interestingly, the FIE arose at a longer exposure than required for above-chance intact-face detection, suggesting that holistic processing is distinct from basic detection of the face as an intact object. Notably, emotional expression did not affect detection: there was a slight, non-significant advantage for neutral faces ($F_{(1, 31)} = 3.633$, $p = 0.066$, $\eta p^2 = 0.105$), with a Bayes Factor indicating strong evidence for the null ($BF_{01} = 10.571$); fearful faces thus conferred no advantage for face detection (see Supplementary Note 3 for the full results of Experiment 1, and Supplementary Fig. 4A for location response bias results).

Emotion identification (sensitivity to the presence of a fearful expression) also increased with exposure duration ($F_{(4.2, 130.9)} = 12.89$, $p < 0.001$, $\eta p^2 = 0.294$), (Fig. 1C; see also Supplementary Fig. 4B for identification criterion results). An effect of face orientation ($F_{(1, 31)} = 19.54$, $p < 0.001$, $\eta p^2 = 0.387$) interacted with exposure duration ($F_{(4.8, 147.2)} = 3.12$, $p = 0.012$, $\eta p^2 = 0.091$): an emotion FIE (a marker of emotion-specific processing[27]) arose at durations of 5.3 ms ($t(31) = 3.563, p = 0.041, d = 0.758, CI = 0.004 − 0.54$) and 6.2 ms ($t(31) = 4.762, p < 0.001, d = 1.013, CI = 0.096 − 0.631$). The emotion FIE thus arises at longer exposures than the location FIE, suggesting that less information is required for holistic face processing than for emotion processing.

We used meta-d′ to assess metacognitive sensitivity[28]—the correspondence between participants' location performance and PAS reports—as a proxy measure of awareness[29] (see Supplementary Note 4 for a detailed description). Metacognitive sensitivity increased with duration ($F_{(3.7, 114.4)} = 12.922$, $p < 0.001$, $\eta p^2 = 0.294$), rising above chance at 1.7 ms for upright faces ($t(31) = 5.56, p < 0.001, d = 0.983, CI = 0.178 − 0.384$) and 2.6 ms for inverted faces ($t(31) = 3.755, p < 0.001, d = 0.664, CI = 0.152 − 0.514$); this is roughly similar to the durations observed for location sensitivity, suggesting that awareness and detection sensitivity arose together (Fig. 1D, see also Supplementary Fig. 4C for metacognitive bias results). A FIE ($F_{(1, 31)} = 6.475$, $p = 0.016$, $\eta p^2 = 0.173$) indicated better metacognitive sensitivity to upright than inverted faces. Similar to location sensitivity, emotion did not affect meta-d′ ($F_{(1, 31)} = 0.11$, $p = 0.742$, $\eta p^2 = 0.004$; $BF_{01} = 13.32$), suggesting that faces' emotional content is not prioritised for awareness. No interactions reached significance (all $p > 0.314$).

Because of the many differences between peripheral and foveal vision[30], Experiment 2 examined whether this pattern of results would

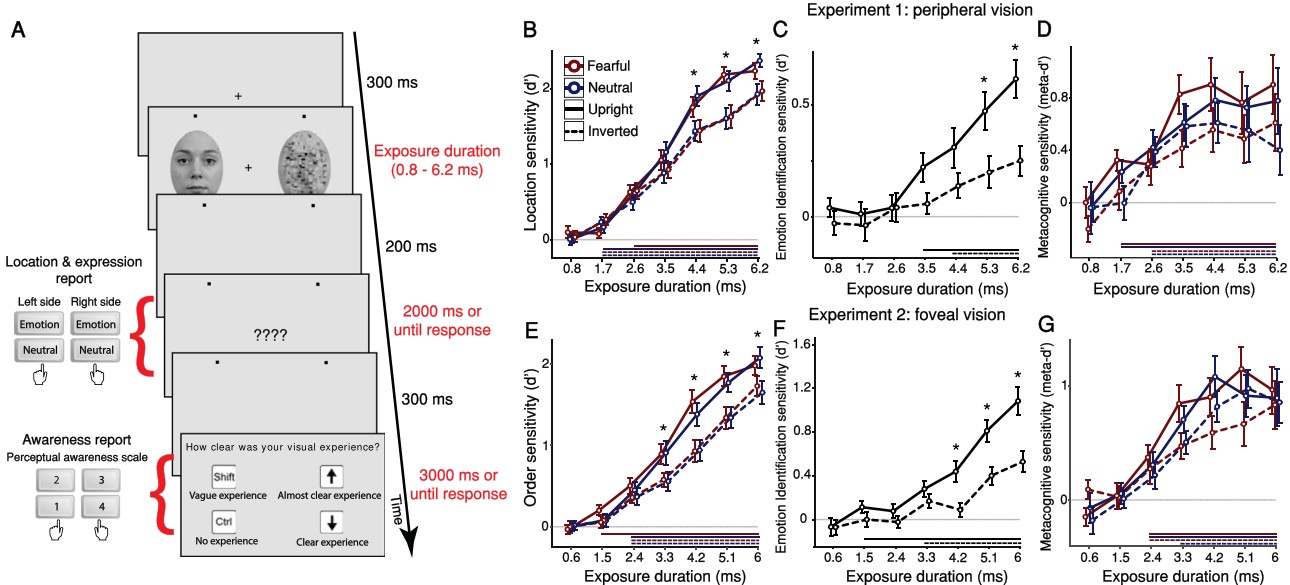

**Fig. 1 | Schematic trial procedure and psychophysical measures of location, emotion identification, and metacognitive sensitivity. A** Experiment 1: An intact and scrambled face were presented for one of seven possible exposure durations (range 0.8–6.2 ms). Participants pressed one key to report both the location (left or right) and expression (emotional or neutral) of the intact face. Next, they reported the clarity of their visual experience. All face stimuli, including the example of an intact face shown here, were taken from the Radboud Face Database (RaFD, see https://rafd.socsci.ru.nl/). **B–D** Findings of Experiment 1 (peripheral vision). **B** Location sensitivity: Two-tailed one-sample t-tests against zero (uncorrected) found that location sensitivity departed from chance-level around 2 ms (all t > 2.469, all $p < 0.003$). A three-way repeated-measures ANOVA (with the factors exposure duration, face orientation, and facial expression) found main effects of exposure duration ($F_{(1.99, 61.97)} = 215.135$, $p < 0.001$, $\eta p^2 = 0.874$), confirming that sensitivity increased with duration, and face orientation ($F_{(1, 31)} = 34.918$, $p < 0.001$, $\eta p^2 = 0.53$), confirming an upright face advantage (face-inversion effect, FIE). Bonferroni-corrected post hoc tests revealed a significant FIE from 4.4 ms of exposure ($t(31) = 5.68$, $p < 0.001$, $d = 0.671$, $CI = 0.125 - 0.65$). There was no main effect of facial expression ($F_{(1, 31)} = 3.633$, $p = 0.066$, $\eta p^2 = 0.105$; $BF_{01} = 10.571$). **C** Emotion identification sensitivity: A two-way repeated-measures ANOVA (with the factors exposure duration and face orientation) found main effects of exposure duration ($F_{(4.2, 130.9)} = 12.89$, $p < 0.001$, $\eta p^2 = 0.294$) and face orientation ($F_{(1, 31)} = 19.54$, $p < 0.001$, $\eta p^2 = 0.387$); Bonferroni-corrected post-hoc tests revealed a significant FIE from 5.3 ms of exposure ($t(31) = 3.563$, $p = 0.041$, $d = 0.758$, $CI = 0.004 - 0.54$). **D** Metacognitive sensitivity: A three-way repeated-measures ANOVA (with the factors exposure duration, face orientation, and facial expression) found main effects of exposure duration ($F_{(3.7, 114.4)} = 12.922$, $p < 0.001$, $\eta p^2 = 0.294$) and face orientation ($F_{(1, 31)} = 6.475$,

$p = 0.016$, $\eta p^2 = 0.173$), suggesting that upright faces required briefer exposures to reach awareness. **E–F** Findings of Experiment 2 (Foveal vision). **E** Order sensitivity: Two-tailed one-sample t-tests against zero (uncorrected) found that location sensitivity departed from chance-level around 2 ms (all t > 3.72, all $p < 0.0005$). A three-way repeated-measures ANOVA (with the factors exposure duration, face orientation, and facial expression) found main effects of exposure duration ($F_{(2.23, 69.02)} = 180.786$, $p < 0.001$, $\eta p^2 = 0.854$) and face orientation ($F_{(1, 31)} = 49.058$, $p < 0.001$, $\eta p^2 = 0.613$). Bonferroni-corrected post hoc tests revealed a significant upright-face (FIE) advantage from 3.3 ms of exposure ($t(31) = 4.737$, $p < 0.001$, $d = 0.584$, $CI = 0.086 - 0.578$). There was no main effect of facial expression ($F_{(1, 31)} = 0.761$, $p = 0.39$, $\eta p^2 = 0.024$; $BF_{01} = 11.891$). **F** Emotion identification sensitivity: A two-way repeated-measures ANOVA (with the factors exposure duration and face orientation) found main effects of exposure duration ($F_{(3.36, 104.12)} = 36.20$, $p < 0.001$, $\eta p^2 = 0.539$) and face orientation ($F_{(1, 31)} = 27.867$, $p < 0.001$, $\eta p^2 = 0.473$); Bonferroni-corrected post-hoc tests revealed a significant FIE from 4.2 ms of exposure ($t(31) = 3.967$, $p = 0.009$, $d = 0.843$, $CI = 0.04 - 0.657$). **G** Metacognitive sensitivity: A three-way repeated-measures ANOVA (with the factors exposure duration, face orientation, and facial expression) found a main effect of face orientation ($F_{(1, 31)} = 6.176$, $p = 0.019$, $\eta p^2 = 0.166$), suggesting that upright faces required briefer exposures to reach awareness. Overall, Experiment 1 and Experiment 2 had very similar results. Horizontal lines below the x-axis of Panels (**B–G**) indicate exposure durations with above-chance sensitivity ($p < 0.05$, one-sample t-test against zero, which is represented by a horizontal grey line). Data are presented as mean values with ±1 SEM bars; $n = 32$ independent participants per experiment. * $p < 0.05$ for upright-inverted comparisons. Source data are provided as a Source Data file.

replicate for fixated stimuli. Thirty-two new observers were shown brief displays of an intact and scrambled face, one after the other, for the same duration. Participants responded as in Experiment 1, but reported whether the intact face was first or second. Because of the better acuity of foveal vision, we set exposure durations to be slightly (0.2 ms) shorter than in Experiment 1. The pattern of results closely replicated that of Experiment 1; the differential effects arose in the same temporal sequence (Figs. 1E–G; see also Supplementary Fig. 5 and Supplementary Note 5 for the full results of Experiment 2). The similarity between the results of Experiments 1 (where faces and scrambles were presented simultaneously) and 2 (where each stimulus was presented alone) indicates that the presence of a low-level-matched scramble alongside the face in Experiment 1 did not interfere with face processing. To assess the possible contribution of afterimages to these observations, we ran two control experiments using reversed-contrast images (which resemble afterimages generated by the main experiments' stimuli), and found that afterimage processing could not

account for the findings (see Supplementary Note 6 and Supplementary Figs. 6–9 for the full results on these control experiments).

Overall, the findings from Experiments 1 and 2 indicate that processing emotion requires longer exposures than holistic face processing (a marker of face-specific processes), which in turn requires longer exposures than intact-face detection. In these experiments, an intact face and its scrambled counterpart were presented on each trial, forcing participants to extract meaning from the stimuli in order to successfully perform the tasks. Next, we asked whether waiving this requirement would lead to the same ordering of processing priorities: Would a face's orientation and expression modulate the ability to detect it, even when it is presented on its own? Detecting a single stimulus can be performed using low-level stimulus attributes (without extracting meaning), reducing the exposure required for localisation; therefore, in Experiment 3 we took advantage of our tachistoscope's ability to display stimuli for sub-millisecond durations

and selected seven new equally-spaced durations (range 0.25–1.25 ms) covering floor to ceiling localisation performance (see Supplementary Fig. 10). The experiment, performed by 32 new participants, had a very similar design to Experiment 1, but with no scrambled faces. As in Experiment 1, location sensitivity increased with duration (Fig. 2A;$F_{(6, 186)} = 300.8$, $p < 0.001$, $\eta p^2 = 0.907$), departing from chance level at 0.417 ms for all categories (fearful upright: $t(31) = 2.689, p = 0.006, d = 0.475, CI = 0.04 - 0.292$; fearful inverted: $t(31) = 3.509, p < 0.001, d = 0.62, CI = 0.092 - 0.349$; neutral upright: $t(31) = 3.558, p < 0.001, d = 0.629, CI = 0.07 - 0.259$; neutral inverted: $t(31) = 1.925, p = 0.032, d = 0.34, CI = -0.006 - 0.222$). Moreover, even without a matched source of low-level noise (scramble), location sensitivity showed a similar pattern to Experiment 1: A main effect of orientation indicated an FIE, with higher location sensitivity for upright than inverted faces ($F_{(1, 31)} = 4.69$, $p = 0.038$, $\eta p^2 = 0.131$); this effect was numerically greater at longer exposure durations, although the interaction between duration and orientation was not significant ($F_{(4.61, 142.9)} = 2.009$, $p = 0.0865$, $\eta p^2 = 0.061$), suggesting that participants' ability to use low-level attributes for localisation may have diluted the sensitivity-enhancement engendered by engaging face-specific mechanisms. Finally, as in Experiment 1, there was no effect of facial expression ($F_{(1, 31)} = 2.031$, $p = 0.164$, $\eta p^2 = 0.061$), nor interaction between expression and duration ($F_{(6, 186)} = 1.21$, $p = 0.303$, $\eta p^2 = 0.038$), indicating no advantage for emotional compared to neutral faces (see Supplementary Note 7 and Supplementary Fig. 11 for more details of Experiment 3).

Unlike Experiment 1, however, these displays were too brief for participants to extract the meaning of facial expressions: identification sensitivity never rose above chance (all $t < 0.781$, $p > 0.22$, $d < 0.4$), and no difference was observed between durations (Fig. 2B; $F_{(6, 186)} = 0.876$, $p = 0.513$, $\eta p^2 = 0.027$). Neither the effect of orientation >($F_{(1, 31)} = 1.324$, $p = 0.259$, $\eta p^2 = 0.041$) nor the interaction ($F_{(4.31, 133.63)} = 0.486$, $p = 0.759$, $\eta p^2 = 0.015$) reached significance. Therefore, we ran Experiment 4 in order to establish the minimal durations required for expression identification in single-stimulus displays. Thirty-two new participants were shown the same stimuli as in Experiment 3 (a single face on each trial, upright or inverted and neutral or fearful), but we used the same seven display durations as in Experiment 1 (0.8–6.2 ms). Under these conditions, unsurprisingly, location sensitivity was above chance for all durations, and at ceiling for all but the shortest duration (Fig. 2C), with no effects of orientation ($F_{(6, 186)} = 1.452$, $p = 0.237$, $\eta p^2 = 0.045$) or expression ($F_{(6, 186)} = 0.004$, $p = 0.953$, $\eta p^2 < 0.01$) and no interactions. For expression identification, we anticipated that performance should be unaffected by the presence or absence of low-level-matched scrambles because this task requires observers to integrate stimulus features and decide which category they belong to. Indeed, observers' ability to identify the expressions in each face closely replicated that seen in Experiment 1 (Fig. 2D). Emotion identification increased with duration ($F_{(3.89, 120.665)} = 17.589$, $p < 0.001$, $\eta p^2 = 0.362$), with an advantage for upright over inverted faces ($F_{(1, 31)} = 15.993$, $p < 0.001$, $\eta p^2 = 0.34$) that interacted with exposure duration ($F_{(6, 186)} = 4.117$, $p < 0.001$, $\eta p^2 = 0.117$), arising at exposures of 5.3 ms ($t(31) = 4.014$, $p = 0.008, d = 0.972, CI = 0.051 - 0.766$) and 6.2 ms ($t(31) = 5.162$, $p < 0.001, d = 1.251, CI = 0.168 - 0.883$). (See Supplementary Note 8 and Supplementary Fig. 12 for more details of Experiment 4). Indeed, emotion identification did not differ between Experiment 1 and 4 ($F_{(1, 62)} = 0.012$, $p = 0.912$, $\eta p^2 < 0.01$), indicating that the presence or absence of scrambled stimuli did not influence emotion identification (see Supplementary Fig. 13 for a direct comparison between Experiments 1 and 4). Overall, the sequence of processing priorities established in Experiments 1 and 2 was replicated in Experiments 3 and 4, using single-stimulus, no-scramble displays. Finally, we note that it was not possible to estimate metacognitive

sensitivity for these data, because participants provided too few high-visibility ratings at all (Experiment 3) or most (Experiment 4) exposure durations, meaning that there was not enough of the necessary variability in scores for model fitting (see Supplementary Note 9 and Supplementary Fig. 14 for details).

Taken together, the findings so far demonstrate an unfolding sequence of processing priorities, indicated by increasing required exposure durations for detection of stimulation, detection of an intact stimulus, holistic face processing, and emotion-specific processing; where calculation of awareness measures was possible, we found that upright faces gain access to awareness faster than inverted faces, whereas emotional expression does not affect either detection or awareness. Importantly, detection and awareness appear to arise together: we find no evidence for unconscious processing, which would be indicated by above-chance sensitivity at shorter exposures for detection than for awareness. This evidence, however, is based on behavioural markers. Might the underlying neural activity unfold in ways that are not revealed by behavioural measures? And could such activity be found at shorter durations than those required for awareness, suggesting unconscious processing?

## Neural markers of minimal required exposures

We used EEG to address these questions in Experiment 5, which specifically aimed to examine markers of emotion processing as facial expressions had no behavioural effects on detection sensitivity: Thirty-two new observers viewed similar displays to those of Experiment 1, but all faces were upright, and in addition to trial blocks with fearful and neutral expressions, we included blocks with happy and neutral expressions (see Supplementary Note 10 for methodological description and Supplementary Figs. 15–16). To increase statistical power, we used just three durations: 1.7 ms (at which, according to Experiment 1, detection and awareness are above chance but holistic face processing and emotion-specific processing are absent), 4.4 ms (when holistic processing arises), and 6.2 ms (when emotion-specific processing is present). We used displays with a face and scramble, as in Experiments 1 and 2; Experiment 4 demonstrated that the absence of a scramble does not alter the required durations for expression identification, and displays with scrambles allowed us to simultaneously measure variation in both location sensitivity and emotion identification.

Psychophysical sensitivity measures increased with duration, and were similar to the same durations' in Experiment 1 (see Supplementary Fig. 17 for psychophysical results): Location sensitivity was already above chance at 1.7 ms, but expression had no effect on it ($F_{(2, 62)} = 0.397, p = 0.674$, $\eta p^2 = 0.013$) and did not interact with duration ($F_{(4, 124)} = 1.711$, $p = 0.152$, $\eta p^2 = 0.052$); emotion identification sensitivity was above chance from 4.4 ms (fearful: $t(31) = 3.113, p < 0.001, d = 0.55, CI = 0.064 - 0.308$; happy: $t(31) = 5.733, p < 0.001, d = 1.014, CI = 0.265 - 0.558$). Expression did not affect metacognitive sensitivity ($F_{(2, 62)} = 1.297$, $p = 0.281$, $\eta p^2 = 0.04$), which was already above chance at 1.7 ms (fearful: $t(31) = 2.843, p = 0.008, d = 0.503, CI = 0.037 - 0.223$; neutral: $t(31) = 3.268, p = 0.003, d = 0.578, CI = 0.068 - 0.293$), but significantly higher (and similar) at 4.4 and 6.2 ms.

We extracted several EEG event-related potentials (ERPs). We focus here on those related to emotion and awareness (for others, see Supplementary Figs. 18–19). Two ERP markers of emotion processing—the early posterior negativity (EPN[31]) and late positive potential (LPP[32])—respond to intensity rather than valence, so fearful and happy expressions were collapsed and compared to neutral ones. For both markers, emotion interacted with duration (EPN: $F_{(2, 62)} = 8.675$, $p < 0.001$, $\eta p^2 = 0.219$; LPP: $F_{(1.98, 61.4)} = 9.804$, $p < 0.001$, $\eta p^2 = 0.24$), showing a significant difference between emotional and neutral faces only at 6.2 ms, the longest exposure (EPN: $t(31) = 4.009, p = 0.002, d = 0.162, CI = -0.791 - -0.112$; LPP: $t(31) = 4.284, p < 0.001, d = 0.142, CI = 0.103 - 0.592$), (Fig. 3;

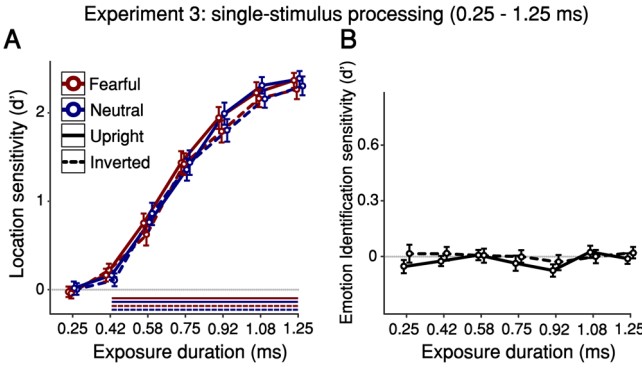

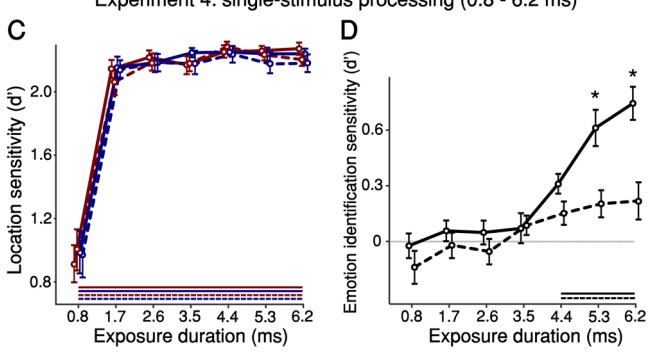

**Fig. 2 | Psychophysical measures of location detection and emotion identification in single-stimulus processing. A**–**B** Findings of **Experiment 3. A** Location sensitivity: Two-tailed one-sample t-tests against zero (uncorrected) found that location sensitivity departed from chance at 0.417 ms (all t > 1.92, all $p < 0.032$). A three-way repeated-measures ANOVA (with the factors exposure duration, face orientation, and facial expression) found a main effect of exposure duration (Fig. 2A; $F_{(6,186)} = 300.8$, $p < 0.001$, $\eta p^2 = 0.907$), confirming that sensitivity increased with duration, and a small but significant main effect of face orientation ($F_{(1,31)} = 4.69$, $p = 0.038$, $\eta p^2 = 0.131$)—an upright-face advantage (FIE). There was no main effect of facial expression ($F_{(1,31)} = 2.031$, $p = 0.164$, $\eta p^2 = 0.061$). **B** Emotion identification sensitivity: Two-tailed one-sample t-tests against zero (uncorrected) revealed that identification sensitivity never departed from chance (all t < 0.781, $p > 0.22$). A two-way repeated-measures ANOVA did not find main effects of exposure duration ($F_{(6,186)} = 0.876$, $p = 0.5134$, $\eta p^2 = 0.027$) or face orientation ($F_{(1,31)} = 1.324$, $p = 0.259$, $\eta p^2 = 0.041$). **C**–**D** Findings of **Experiment 4. C** Location sensitivity: Two-tailed one-sample t-tests against zero (uncorrected) showed that location sensitivity was above chance at all durations (all t > 6.84, all $p < 0.001$), reaching ceiling from 1.7 ms of exposure. A three-way repeated-measures ANOVA (with the factors exposure duration, face orientation, and facial expression) showed only a main effect of exposure duration ($F_{(1.276, 39.557)} = 143.034$, $p < 0.001$, $\eta p^2 = 0.822$). **D** Emotion identification sensitivity: A two-way repeated-measures ANOVA showed main effects of exposure duration ($F_{(3.89, 120.665)} = 17.589$, $p < 0.001$, $\eta p^2 = 0.362$) and face orientation ($F_{(1,31)} = 15.993$, $p < 0.001$, $\eta p^2 = 0.34$). Bonferroni-corrected post hoc tests revealed a FIE arising from 5.3 ms of exposure ($t(31) = 4.014$, $p = 0.008$, $d = 0.972$, CI = 0.051 − 0.766). These results closely replicated the identification sensitivity results of Experiment 1. Horizontal lines below the x-axes represent above-chance sensitivity ($p < 0.05$, one-sample t-test against zero, which is represented by a horizontal grey line). Data are presented as mean values with ±1 SEM bars; $n = 32$ independent participants per experiment. * $p < 0.05$ for upright-inverted comparisons. Source data are provided as a Source Data file.

see Supplementary Fig. 20 for additional topographies and analyses of these ERP markers). These ERP findings suggest, in line with our psychophysical results, that the minimal exposure required for emotion-specific processing is between 4.4 and 6.2 ms (see Supplementary Note 11 for the full results of Experiment 5).

Because ERP analysis may not be sufficiently sensitive to neural patterns across electrodes, we examined emotion-related processing with multivariate pattern analysis (MVPA[33]). First, we trained a classifier to decode the location of the intact face. The classifier successfully decoded face location at 4.4 and 6.2 ms of exposure (Figs. 4A–F). However, no differences in decoding accuracy were found between expressions, suggesting that no expression is prioritised for processing (Figs. 4G–I). Next, we trained a classifier to decode the 3 expressions, regardless of their location. The classifier had limited success at 4.4 ms, and robust success at 6.2 ms (Figs. 4J–K), demonstrating somewhat better sensitivity than ERPs but supporting the overall conclusion that emotion-specific processing requires at least 4 ms exposure.

To examine whether processing meaningful stimuli requires awareness, we divided trials into those with a subjective report of no awareness ("no experience" in the PAS) and those in which there was some level of awareness (all other PAS responses). Comparing the two categories, we extracted two ERP markers of awareness: the visual awareness negativity (VAN[34]) and the late positivity (LP[35,36]). As there were not enough no-awareness trials at 6.2 ms, this analysis was limited to durations of 1.7 and 4.4 ms (see Supplementary Fig. 21). For both markers, there was an interaction between duration and awareness (VAN: $F_{(1, 30)} = 10.062$, $p = 0.003$, $\eta p^2 = 0.251$; LP: $F_{(1, 30)} = 37.42$, $p < 0.001$, $\eta p^2 = 0.555$); both ERPs distinguished between presence and absence of awareness significantly only at 4.4 ms (VAN: $t(45.4) = 5.205$, $p < 0.001$, $d = 0.327$, CI : $−1.632 − −0.501$; LP : $t(55) = 5.861$, $p < 0.001$, $d = 0.56$, CI : $0.622 − 1.713$), (Fig. 5); for VAN (but not LP), there was a numerical trend at 1.7 ms, in line with the small but above-chance meta-d' at this duration (see Supplementary Fig. 22 for left and right VAN). Thus, neural findings show that the minimal exposure for awareness is equivalent to or precedes that for emotion processing.

Do neural markers require similar visual exposures to those found above in order to discriminate faces from non-face stimuli? In Experiment 6, 32 new observers viewed similar displays to those of Experiment 5, but each consisted of either a neutral-expression face or a highly-recognisable object (and their scrambled counterpart; see Supplementary Note 12). We used four display durations (range 0.8–4.288 ms). Participants reported the location (left/right) and category (face/object) of the intact image, followed by a PAS rating.

Psychophysically, the minimal duration required for above-chance location and identification sensitivity was 1.4 ms (faces: $t(31) = 2.65$, $p = 0.013$, $d = 0.468$, CI = 0.026 − 0.198; objects: $t(31) = 4.076$, $p < 0.001$, $d = 0.721$, CI = 0.072 − 0.217; Identification: $t(31) = 2.09$, $p = 0.045$, $d = 0.369$, CI = 0.002 − 0.19). Meta-d' was not affected by stimulus category ($F_{(1, 31)} = 0.902$, $p = 0.35$, $\eta p^2 = 0.028$, $BF_{01} = 4.67$) and was above chance at 1.4 ms for both faces ($t(31) = 2.3$, $p = 0.028$, $d = 0.407$, CI = 0.016 − 0.262) and objects ($t(31) = 3.64$, $p < 0.001$, $d = 0.644$, CI = 0.103 − 0.364), (see Supplementary Fig. 23 for psychophysics results).

We used an ERP marker of face processing, the N170 (which is greater for faces than objects[8,37]) to assess the minimal duration for specific face-related processing (see Supplementary Fig. 24 for the P1, a marker of early visual processing). The effects of duration and category on amplitude interacted ($F_{(2.5, 75.9)} = 6.398$, $p = 0.001$, $\eta p^2 = 0.171$): the N170 successfully discriminated faces from objects only at 4.288 ms ($t(31) = 3.467$, $p = 0.021$, $d = 0.134$, CI: $−0.668 − 0.027$), (Fig. 6). These neural findings show a similar minimal exposure for face-specific processing—around 4 ms—as that needed for the FIE to arise in Experiment 1, converging on this duration as the minimal exposure required for engaging face-specific processes (see Supplementary Fig. 25 for additional topographies of the N170 and Supplementary Fig. 26 for an exploratory analysis on late latencies; see Supplementary Fig. 27 for VAN results and Supplementary Fig. 28 for LP results; and see Supplementary Note 12 for the full results of Experiment 6).

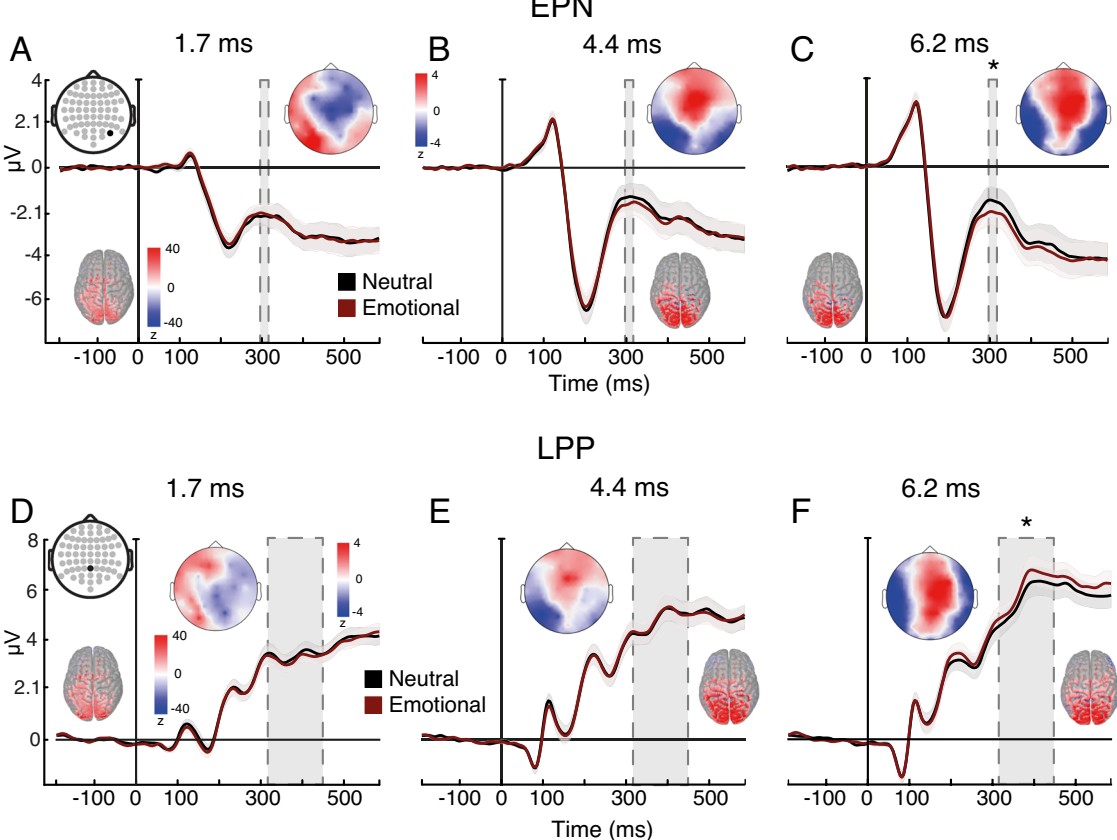

**Fig. 3 | Experiment 5: Neural measures of emotion processing. A–C** EPN marker of emotion-processing across exposure durations. A three-way repeated-measures ANOVA (with the factors exposure duration, facial expression, and brain hemisphere) found an interaction between facial expression and exposure duration ($F_{(2,62)} = 8.675, p < 0.001, \eta p^2 = 0.219$). Bonferroni-corrected post-hoc tests revealed that the evoked response to emotional expressions was significantly more negative than to neutral expressions, indicating emotion-specific processing, only at 6.2 ms of exposure ($t(31) = 4.009, p = 0.002, d = 0.162, CI = -0.791 - -0.112$). **D–F** LPP marker of emotion-processing across exposure durations. A two-way repeated measures ANOVA (with the factors exposure duration and facial expression) found an interaction between facial expression and exposure duration ($F_{(1.98, 61.4)} = 9.804, p < 0.001, \eta p^2 = 0.24$). Bonferroni-corrected post-hoc tests revealed that the evoked response to emotional expressions was significantly more positive than to neutral expressions, indicating emotion-specific processing, only at 6.2 ms of exposure ($t(31) = 4.284, p < 0.001, d = 0.142, CI = 0.103 - 0.592$). Topographic maps represent emotional−neutral voltage subtraction in Z-scores. Source estimation of the ERPs at their peaks are shown on cortical maps. Time in all x-axes is from stimulus onset. Data are presented as mean values with ±1 SEM bars; n = 32 independent participants. * p < 0.05 for emotion-neutral comparisons. Source data are provided as a Source Data file.

## Discussion

Overall, psychophysical and neural measures converge to clarify the sequence of minimal exposures (i.e., minimal bottom-up stimulation) needed for face processing to unfold: increasing exposure durations are required for above-chance stimulus detection, intact-stimulus detection, face-specific processing, and emotion-specific processing. Awareness arises hand-in-hand with detection, increasing with exposure duration. Face orientation affects the minimal exposures required for detection and awareness, indicating prioritisation of holistic processing. Emotion processing, however, does not affect minimal exposures, and is only evident at longer durations than those at which face detection is reliable and face-specific processes are engaged. No sensitivity measures rise above chance at shorter exposures than required for awareness.

These findings set upper bounds on the minimal exposures required for stimulus detection, object detection, face-specific processing, emotion processing, and visual awareness: Some of the specific durations appear to be paradigm-dependent (for example, the FIE on localisation arose at around 1 ms for single-stimulus displays, but required around 3–4 ms for face + scramble displays), whereas others are not (expression identification required around 5 ms regardless of display type). Crucially, although shorter durations may yet be found for higher-luminance or more distinctive stimuli (see Supplementary Note 13 for a discussion), longer minima are ruled out. Interestingly,

our findings demonstrate that seeing a face as an intact object is not sufficient for engaging face-specific processing, and that the presence of face-specific processing does not guarantee emotion processing. This contradicts suggestions that perceiving an object's location entails perceiving its full nature[38]; furthermore, it is inconsistent with the proposal that awareness of all stimulus attributes arises simultaneously (e.g., through 'ignition'[9,10]). Our measure of awareness only assessed subjective experience in general, rather than awareness of specific attributes; but the presence of above-chance metacognitive sensitivity at shorter durations than those required for holistic or emotion processing suggests that observers have some awareness of the intact stimulus before perceiving those attributes, supporting the possibility that awareness of different aspects of the same stimulus may require different amounts of information.

While our findings lay out a sequence of processing priorities in the visual system, they leave open the precise causes of this sequence. To some degree, we should expect the visual system's priorities to be a simple reflection of lower-level stimulus characteristics, such as energy −visual features that require less energy for discrimination are effectively given higher priority. But in other respects, the measured priorities clearly indicate that human vision is adapted to extract meaning from specific types of information over others. Upright faces, for example, were clearly prioritised over inverted faces, with shorter

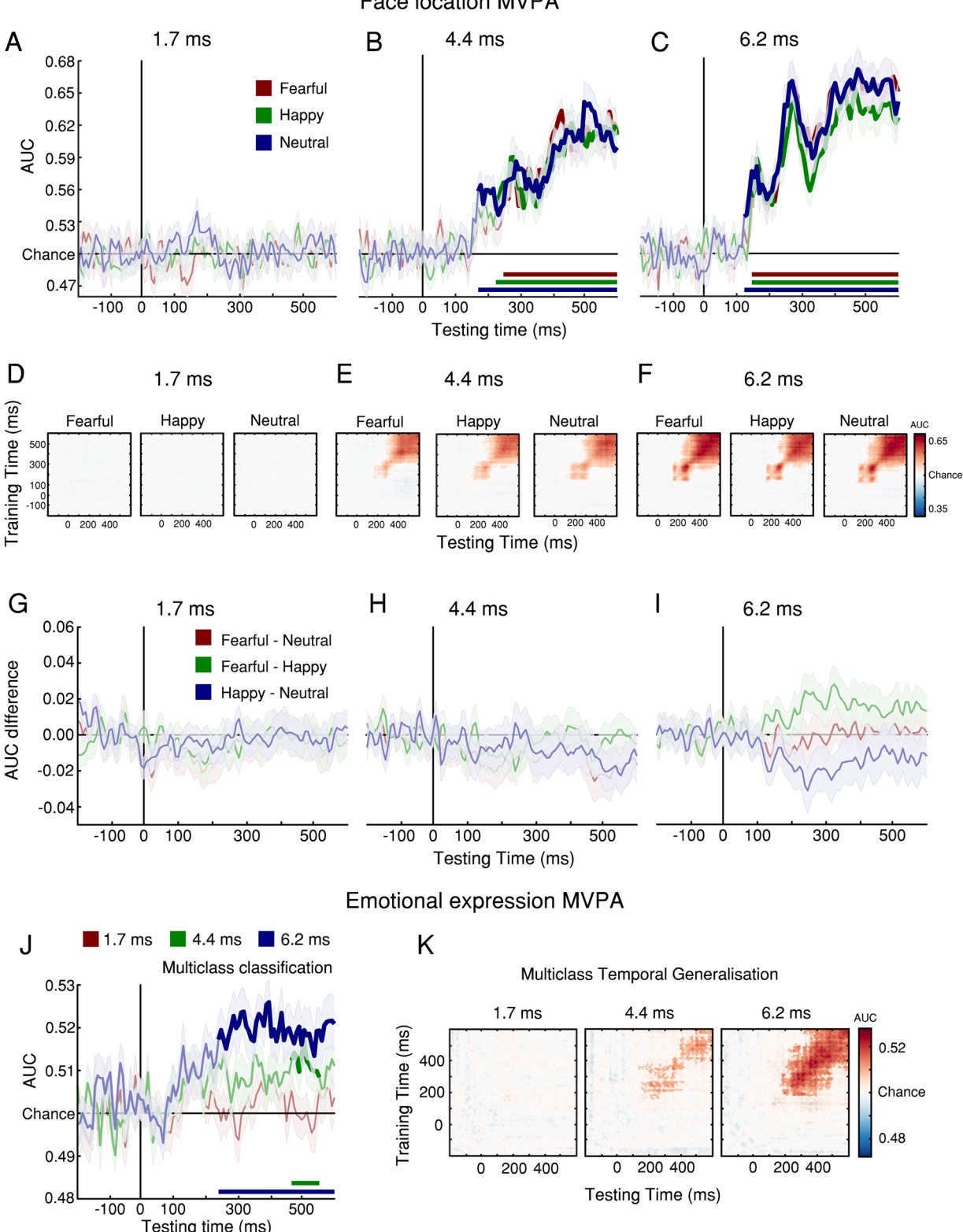

exposures required to detect the former than the latter, even though stimulus energy was identical between the two. Conversely, differences in emotion expression did not, in fact, influence minimal required exposures even though their stimulus energy did differ, suggesting that contrary to prior reports[21,39,40], the visual system is not attuned to certain types of emotional expression.

Finally, the absence of any evidence for subliminal perception reveals awareness primacy: Although masking studies have provided evidence for unconscious processing[21,41–43], our findings suggest that when not actively disrupted by masks, perceptual processing and awareness arise together. This raises the possibility that awareness may be a condition for certain types of processing.

**Fig. 4 | Experiment 5: MVPA decoding of face location and emotional expression. A–G** MVPA of intact-face location. **A–C** Classifier performance (in units of area under curve, AUC) shows that the location of intact faces was decoded significantly above chance only at 4.4 and 6.2 ms of exposure. Classification accuracy was calculated by comparing participants' AUC scores against 0.5 (chance performance) through t-tests using cluster-based permutation testing. Bold segments represent significant clusters ($p < 0.05$, cluster-corrected). **D–F** Temporal generalisation analysis. The Y-axis depicts training time points, and the X-axis depicts testing time points, relative to stimulus presentation (time zero). AUC scores revealed broad temporal generalisation of the decoded multivariate patterns at 4.4 and 6.2 ms of exposure. **G–I** AUC difference between pairs of expressions. No paired comparison revealed significant clusters, suggesting that no emotional expression enjoyed above-chance classification at any exposure duration. **J** Multiclass MVPA decoding of expression showed limited success for 4.4 ms and robust classification for 6.2 ms exposure. Bold segments represent significant clusters ($p < 0.05$, cluster-corrected). **K** Temporal generalisation shows cortical signal stability across time. Solid bold lines at the bottom of the charts represent the times of significant clusters ($p < 0.05$, cluster-corrected) and shaded contours represent ±1 SEM; $n = 32$ independent participants. Source data are provided as a Source Data file.

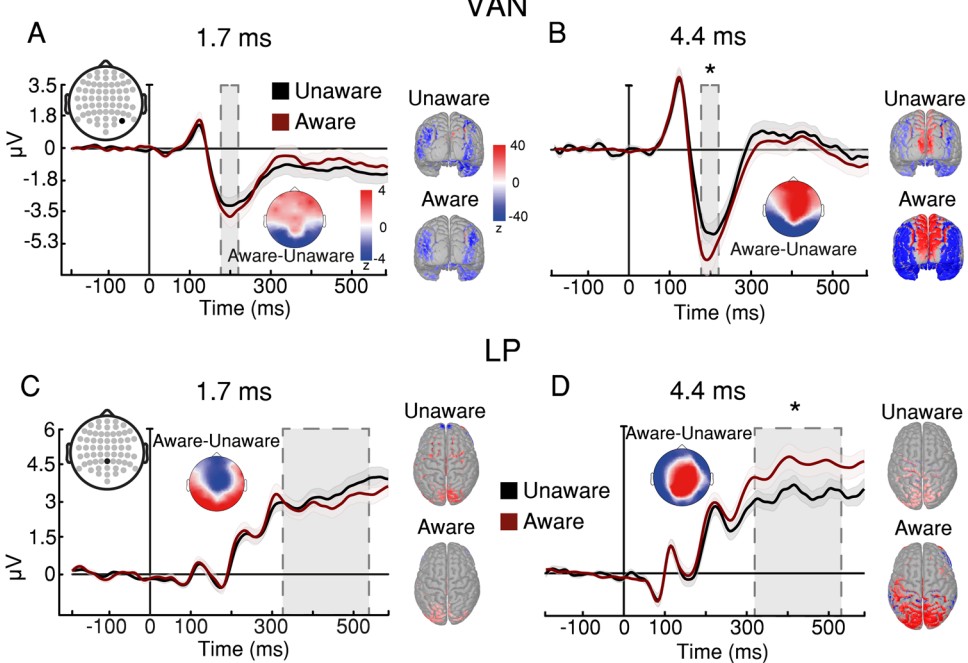

**Fig. 5 | Experiment 5: Neural measures of awareness. A–B** VAN marker of awareness across exposure durations. A four-way repeated-measures ANOVA (with the factors exposure duration, awareness report, facial expression, and brain hemisphere) found an interaction between awareness report and exposure duration ($F_{(1,30)} = 10.062$, $p = 0.003$, $\eta p^2 = 0.251$). Bonferroni-corrected post-hoc tests revealed that the evoked response in awareness-present trials was significantly more negative than in awareness-absent trials only at 4.4 ms of exposure ($t(45.4) = 5.205$, $p < 0.001$, $d = 0.327$, CI : $-1.632 - -0.501$). **C–D** LP marker of awareness across exposure durations. A three-way repeated-measures ANOVA (with the factors exposure duration, awareness report, and facial expression) found an interaction between awareness report and exposure duration ($F_{(1,30)} = 37.420$, $p < 0.001$, $\eta p^2 = 0.555$). Bonferroni-corrected post-hoc tests revealed that the evoked response in awareness-present trials was significantly more positive than in awareness-absent trials only at 4.4 ms of exposure ($t(55) = 5.861$, $p < 0.001$, $d = 0.56$, CI : $0.622 - 1.713$). Time in all x-axes is from stimulus onset. Shaded contours represent ±1 SEM. * $p < 0.05$ for aware-unaware comparisons; $n = 31$ independent participants. Source data are provided as a Source Data file.

Great efforts have been made to characterise the visual processing hierarchy, mostly by describing the anatomical arrangement of specialised processing regions[44,45]. The present study offers a complementary approach, enabling elucidation of the functional dynamics of this hierarchy—how well the system is attuned to different stimulus attributes—by assessing the minimal exposure required for the relevant information to propagate through the system and reach dedicated processing. The face stimuli used here provide a proof of concept for the utility of this approach, which may yield further valuable insights by application to both lower-level features and higher semantic categories.

## Methods

### Participants

The experiments were approved by the Université libre de Bruxelles Faculty of Psychological Science and Education ethics committee. Participants gave informed consent and received €15 for participation in Experiment 1 through 4, and €30 for participation in Experiment 5 and Experiment 6. In Experiment 1, 35 participants were recruited – 3

were excluded (2 failed to provide a response on more than 5% of the trials and 1 had chance accuracy in all exposure duration conditions), leaving a sample of 32 participants (M_age = 22.6 [SD_age = 2.7]; 17 female). In Experiment 2, 34 participants were recruited – 2 were excluded (they failed to provide a response on more than 5% of the trials), leaving a sample of 32 participants (24.6 [5.5]; 19 female). In Experiment 3, 34 participants were recruited – 2 were excluded (they failed to provide a response on more than 5% of trials), leaving a sample of 32 participants (25.5 [3.8]; 20 female). In Experiment 4, 33 participants were recruited – 1 was excluded (they failed to provide a response on more than 5% of trials), leaving a sample of 32 participants (24.8 [5.1]; 20 female). In Experiment 5, 36 participants were recruited – 4 were excluded (they presented more than 15% of noisy electrodes during EEG signal pre-processing), leaving a sample of 32 participants (24.3 [4.9]; 18 female). One additional participant was excluded from the analysis of VAN and LP for not providing PAS ratings of 'almost clear experience' and 'clear experience.' In Experiment 6, 38 participants were recruited – 6 were excluded (they presented more than 15% of noisy electrodes during EEG signal pre-processing), leaving a sample

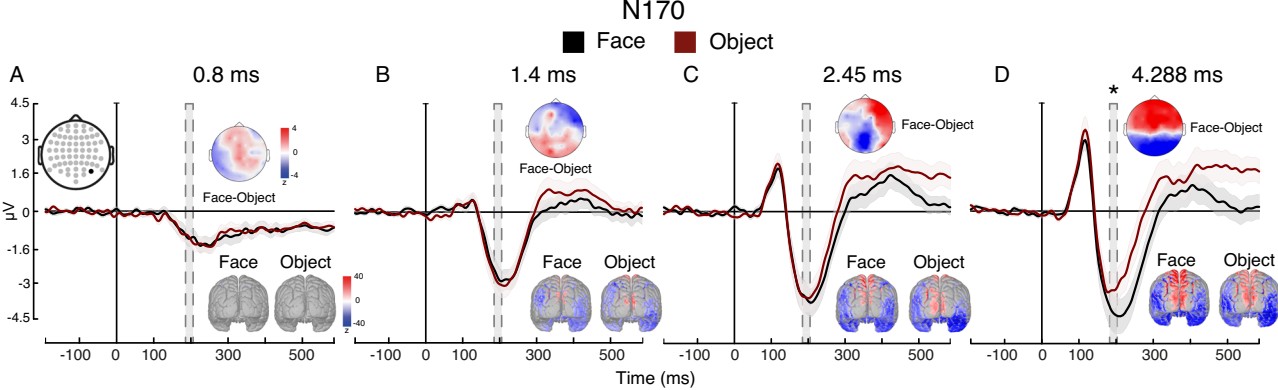

**Fig. 6 | Experiment 6: Neural measures of face vs. object processing.**
**A**–**D** Findings for each of the four durations. A three-way repeated-measures ANOVA (with the factors exposure duration, facial expression, and electrode) found an interaction between stimulus category and exposure duration ($F_{(2.45, 75.89)} = 6.398, p = 0.001, \eta p^2 = 0.171$). Bonferroni-corrected post-hoc tests revealed that the evoked response to face stimuli was significantly more negative

than to object stimuli only at 4288 ms of exposure ($t(31) = 3.467, p = 0.021$, $d = 0.134$, CI : $-0.668 - -0.027$). Topographic maps represent face−object voltage subtraction in Z-scores. Source estimation of the ERPs at their peaks are shown on cortical maps. Time in all x-axes is from stimulus onset. Shaded contours represent ±1 SEM. * $p < 0.05$ for face-object comparisons; $n = 32$ independent participants. Source data are provided as a Source Data file.

of 32 participants (23.2 [3.2]; 21 female). One additional participant was excluded from the analysis of VAN and LP for not providing PAS ratings of 'almost clear experience' and 'clear experience.' Participants self-reported their gender and age; however, this information was not considered in the study design or included in the data analysis because we had no hypothesis involving it. All participants had normal or corrected-to-normal vision and reported no history of neurological or psychiatric disorders.

### Display and apparatus
The custom-made LCD tachistoscope used in the study is based on a design described by Sperdin et al.[14,46]. Two LCD screens are employed: one vertical and the other horizontal, aligned to the top of the vertical screen. A semi-permeable mirror is placed diagonally between the two screens, allowing light to pass from the vertical screen and reflecting light from the horizontal screen. Both screens are thus visible to the observer, superimposed on each other when their backlights are simultaneously on. The setup enables control of which screen is visible to the observer, by controlling the screens' backlights, which are controlled by a dedicated micro-controller enabling precision of $0.002 \pm 0.001$ ms. In all our experiments, the vertical screen displayed all images that were not shown for durations shorter than 10 ms (fixation, placeholders, response cue, etc.), and its backlight was therefore always on. The horizontal screen displayed the stimuli that were shown for durations shorter than 10 ms, and its backlight was therefore off, except during those brief stimulus presentations; during these periods, the content of both screens was visible to the observer. For full technical details of the apparatus, see Supplementary Note 1.

### Psychophysics
We used Signal Detection Theoretic (SDT) measures to assess how perceptual sensitivity, metacognitive sensitivity, and decision criteria changed across display durations. To determine each participant's bias-independent sensitivity to face location (left or right; henceforth referred to as location d') for each combination of duration, face orientation, and emotional expression, we employed the calculation for two-alternative forced choice (2AFC) tasks for perceptual sensitivity and the criterion (type-1 SDT[47]), $d'_{location}=(1/\sqrt{2})(Z(Hit_{location})-Z(FA_{location}))$, where Z(Hit) stands for the Z score associated with the probability of a Hit (defined as a trial in which a face was displayed on the right and reported on the right), and Z(FA) for that associated with the probability of a false alarm (a trial in which a face was displayed on

the left but reported as being on the right). To estimate each participant's bias to respond left or right (henceforth referred to as response bias) during face location, we employed the calculation $C_{location} = -\left(\frac{1}{2}\right)\left(Z(Hit_{location}) + Z(FA_{location})\right)$. Positive and negative values for this measure indicate a bias toward responding "left" and "right", respectively; however, as these may cancel out across participants, we converted the results to absolute values as a measure of response bias quantity. To determine how sensitive each participant's awareness judgement (PAS ratings) was to their location sensitivity performance (metacognitive sensitivity; henceforth referred to as meta-d'), and the bias in such judgements (metacognitive bias; henceforth referred to as meta-bias), we employed the maximum likelihood estimation procedure developed by Maniscalco and Lau[28,48,49] (http://www.columbia.edu/~bsm2105/type2sdt/).

To determine emotional identification sensitivity (to fearful expressions versus neutral expressions; henceforth referred to as identification d'), we used the calculation of d' for Yes-No detection tasks, $d'_{identification} = Z(Hit_{identification}) - Z(FA_{identification})$, where a hit was defined as correctly reporting a fearful expression and FA was defined as incorrectly reporting a fearful expression. To estimate each participant's identification criterion during emotion expression identification, we employed the calculation $C_{identification} = -\left(\frac{1}{2}\right)\left(Z(Hit_{identification}) + Z(FA_{identification})\right)$. These measures were calculated by-condition for each participant and analysed using analysis of variance (ANOVA). Greenhouse-Geisser adjusted degrees of freedom were used when Mauchly's test indicated a violation of the sphericity assumption.

### Electroencephalography
EEG data were recorded and digitised at a sampling rate of 512 Hz using a 64-channel Biosemi system with an elastic cap, in which electrodes were integrated at sites conforming to the 10-20 system. All impedance values were kept below 50 kΩ. Scalp electrodes were referenced to Cz. The continuous EEG data was resampled to 256 Hz, then filtered leaving frequencies between 0.3 and 40 Hz, and finally epoched from 200 ms before to 600 ms after stimulus onset. An independent-component analysis (ICA) was run on the epoched EEG signal. Components attributed to eye blinks, ocular movements, heartbeat, and channel noise were taken out. Trials with voltage exceeding 150 mV were excluded from further analysis. On average, 1.8% of trials were removed. The EEG signal was then re-referenced to the average across all electrodes. Waveforms were then averaged for all electrodes. By eye

inspection on canonical sites, we determined the following electrodes for each event-related potential (ERP) component of interest: P1 (Oz), left N170 (PO7), right N170 (PO8), left VAN (PO7), right VAN (PO8), VPP (FCz), left EPN (P7), right EPN (P8), LPP (Pz), and LP (Pz). Because emotion processing-related components (EPN and LPP) mainly respond to emotional intensity rather than valence[32,50], and because we wanted to preserve a balanced number of trials between conditions, we collapsed all components across trials, separately for emotional trials (fearful and happy expressions together) and neutral trials (neutral expressions) in Experiment 5. ERP components were collapsed across trials from the same stimulus condition regardless of behavioural performance in the task, except for our pre-registered analyses of VAN and LP (https://aspredicted.org/53sv5.pdf). These two components were collapsed across two groups of trials according to the trials' PAS ratings: awareness-present trials (PAS ratings of "vague impression", "almost clear experience", or "clear experience") and awareness-absent trials (PAS rating of "no experience"). To investigate changes in ERP components, mean amplitudes were computed within the following time windows: 105-135 ms (P1), 170-200 ms (N170 and VPP), 295-325 ms (EPN), 310-430 ms (LPP), 185-215 ms (VAN), and 320-520 ms (LP). These time windows were determined by eye inspection on the grand average plots, in specific time windows informed by previous studies, and did not involve exploratory statistical testing, as a way to reduce familywise error[51,52].

In Experiment 6, EEG recording and pre-processing was performed with two differences: first, based on eye inspection of canonical sites, we selected slightly different time windows for the following ERP components: P1 (105-135), N170/VPP (175–205 ms), VAN (200–230 ms), and LP (300–585 ms). Second, mean amplitudes were not computed for EPN and LPP.

### Event-related potential analysis

In Experiment 5, ERP analysis was performed on mean amplitude values of each ERP component at the specific electrodes and time windows stated above. We reconstructed each ERP's cortical source using Brainstorm (Tadel et al.[53], version released in October 2019). To estimate the cortical source of an ERP component, we need to model the electromagnetic properties of the head and of the sensor array (forward model), and then estimate the brain sources that produced the EEG signal of interest (inverse model). The forward model was calculated using the OpenMEEG Boundary Element Method[54] on the cortical surface of a template MNI brain (ICBM152) with 1 mm resolution. The inverse model was constrained using weighted minimum-norm estimation[55] (wMNE) to measure source activation in picoampere-metres. wMNE looks for a distribution of sources with the minimum current that can account for the EEG data. We corrected grand-averaged activation values by subtracting the mean of the baseline period (−200 to 0 ms before stimulus onset) and spatially smoothed with a 5-mm kernel. This procedure was applied separately for each ERP component.

### Multivariate pattern analysis

To complement ERP analysis, an MVPA on the raw EEG data was applied using the ADAM toolbox[56]. To achieve this, we used a classification algorithm based on linear discriminant analysis (LDA). We used K-fold cross-validation, whereby each participant's dataset was sorted into 10 folds; the classifier was trained on 9 folds and tested on the remaining one. Therefore, training and testing steps were independent from each other. To keep a balanced number of trials between conditions[57], we randomly selected and discarded trials when necessary ("undersampling"[56]). To measure classification performance, we calculated the area under the curve accuracy metric (AUC) of the receiver operating characteristic (ROC), a measure derived from SDT insensitive to classifier bias[58,59]. Finding above-chance classification

performance indicates that there is information contained in the EEG data that was decoded based on the experimental conditions of interest.

Two independent backward decoding algorithms (automated classifiers), following a 10-fold cross-validation approach (including all electrodes and time points), were applied using: (1) intact-face location as class; and (2) emotional expressions as class, i.e., fearful, happy, and neutral expressions (Experiment 5). AUC scores were tested per time point with double-sided t-tests cluster-corrected for multiple comparisons[60], with a standard cut-off p-value of 0.05 (1000-iteration permutation tests).

### Statistical information

Both frequentist (repeated-measures ANOVA and two-tailed t-tests) and Bayesian (Bayes factors) statistical analyses were performed JASP[61] and corroborated using R. When an ANOVA indicated a significant interaction, we ran post hoc Bonferroni-corrected pairwise comparisons to look for significant effects. Post hoc pairwise comparisons in these statistical packages use estimated marginal means based on the variance of the ANOVA model. For Bayes factor analysis, we defined the null hypothesis as no difference between conditions by using a standard Cauchy prior distribution centred around zero with a width parameter of 0.707.

### Reporting summary

Further information on research design is available in the Nature Portfolio Reporting Summary linked to this article.

### Data availability

The aggregated psychophysical and pre-processed EEG data generated in this study have been deposited in the Open Science Framework (OSF). The data used to create the figures are shared in the Source Data file. Raw data will be available upon request starting in January 2027. For access, please contact Dr. Renzo Lanfranco (Renzo.Lanfranco@ki.se). Source data are provided with this paper.

### Code availability

Custom code used to extract individual participant behavioural results and custom code used to pre-process electroencephalographic signal are available in the OSF.

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

## Acknowledgements

R.C.L., H.R., and D.C. thank Dr. Simon van Gaal and Prof. Robert McIntosh for their valuable comments and suggestions. R.C.L. thanks Dr. Johannes Fahrenfort, Dr. Maximilian Bruchmann, Prof. Stephen Fleming, Dr. Albert De Beir, and Dr. Brian Maniscalco for their useful technical advice. The authors thank Dr. Dalila Achoui for her technical guidance. R.C.L. was supported by an ANID/CONICYT PhD studentship, a British Psychological Society Postgraduate Study Visit Scheme award, a University of Edinburgh Principal's Go Abroad Fund award, a Université libre de Bruxelles Seal of Excellence research fellowship, a Karolinska Institutet Strategic Research Area Neuroscience (StratNeuro) postdoctoral fellowship, and partially by a Karolinska Institutet postdoctoral stipend. A.C-J. was supported by an ANID/FONDECYT Regular (1240899) research grant. H.R. was supported by an Economic and Social Research Council (ESRC) future research leaders award (no. ES/L01064X/1). A.C. was supported by a European Research Council (ERC) advanced grant (EXPERIENCE – ADG101055060). D.C. was supported by an ERC advanced grant (X-SPECT – DLV692739).

## Author contributions

R.C.L., H.R., and D.C. conceived the study. R.C.L., H.R., A.C., and D.C. conceptualised the study. R.C.L., H.R., A.C., and D.C. acquired funding. R.C.L, H.R., and D.C. designed the experiments. R.C.L. programmed the experiments, collected the data, analysed the data, and created the figures. A.C-J., H.R., A.C., and D.C. supervised the data analysis. R.C.L. and D.C. wrote the manuscript with input from all authors.

## Funding

## Competing interests

The authors declare no competing interests.
