## [Peer Review File · Nature Communications]

Editorial Note: This manuscript has been previously reviewed at another journal that is not operating a transparent peer review scheme. This document only contains reviewer comments and rebuttal letters for versions considered at Nature Communications. Mentions of prior referee reports have been redacted.

Response to reviewer comments

1. Reviewer comments are given in full. For clarity, we have reproduced them in **blue** font. Any new or substantially altered text is highlighted in **yellow**, both in the responses below and in the manuscript files.

2. Reviewer comments are from the last (second) round of reviews. [REDACTED]

At the [REDACTED]editor's suggestion, the reviews from both rounds, and our responses to the first round, have been shared with Nature Communications.

[REDACTED]

Response:

We acknowledge the reviewer's apprehension, the necessity for additional clarifications, and most importantly, the requirement for empirical evidence to demonstrate that our main findings (those concerning the sequence of minimal exposures indicating processing priorities) generalise to single-stimulus displays. Although we maintain that our original design choices (based on established psychophysical methodology that prevents performance relying on low-level attributes) were a strength of our study, we have taken the reviewer's concerns very seriously: Following the reviewer's suggestion, we have added to the manuscript two new experiments that used only a single face image on each trial, with no scrambled stimuli.

Our new experiments are integrated into the manuscript as Experiments 3 and 4. Both are built on our Experiment 1: Participants were presented with the same facial stimuli as in that experiment, and except for the absence of scrambled images, the design was similar (i.e., we manipulated face orientation (upright, inverted), expression (fearful, neutral), and exposure duration (comprising seven different levels)).

Experiment 3

In the absence of scrambles, which provided low-level-matched noise, participants were able to use low-level attributes to detect the location of single stimuli, making this task much easier. Therefore, our first new experiment used a set of briefer presentation durations. To choose these durations, we ran a pilot study where six participants detected the location of neutral upright faces presented for a range of very short durations (0.1–1 ms in 0.05ms increments, and 1–1.6 ms in 0.1 ms increments; 20 trials/duration). We used these data to choose seven new equally-spaced presentation durations for Experiment 3, capturing the range of performance from floor to ceiling (0.250 ms, 0.417 ms, 0.583 ms, 0.750 ms, 0.917 ms, 1.083 ms, and 1.250 ms). The pilot is described in **Supplementary Note 7** and **Supplementary Fig. 10**, quoted below:

“Detecting a single stimulus can be performed using low-level stimulus attributes (without extracting meaning), reducing the exposure required for localisation. Therefore, we took advantage of our LCD tachistoscope's ability to display stimuli for sub-millisecond durations and selected seven new equally-spaced durations (range 0.25 – 1.25 ms) covering floor to ceiling localisation performance. To choose these durations, we ran a pilot study where six participants detected the location of neutral upright faces presented for a range of very short durations (0.1–1 ms in 0.05ms increments, and 1–1.6 ms in 0.1 ms increments; 20 trials/duration). We used the results shown in **Supplementary Fig. 10** to choose seven new equally-spaced presentation durations for Experiment 3, capturing the range of performance from floor to ceiling (0.250, 0.417, 0.583, 0.750, 0.917, 1.083, and 1.250 ms of exposure).”

Supplementary Fig. 10. Pilot results for Experiment 3 (n = 6). Location sensitivity increased from chance level to high sensitivity across exposure durations. Error bars: ± 1 SEM.

The results of Experiment 3 (n=32) are reported in **Lines 200-230** of the revised manuscript, and in **Fig. 2A and 2B** (new text and figures quoted below). For location detection, the findings showed a very similar pattern to that of Experiment 1: Sensitivity increased with exposure duration, facial expression had no effect, and upright faces enjoyed an advantage over inverted faces (a face inversion effect, FIE). The FIE was numerically evident in two of the longer durations (0.917 and 1.083 ms), although the interaction between exposure duration and orientation did not reach significance (suggesting that although engaging face-specific mechanisms enhances sensitivity, this effect may have been diluted by participants' use of low-level feature detection in this single-stimulus task). **These results address the reviewer's concerns by replicating the primary findings of our original Experiment 1 – specifically, by showing that the same hierarchy of visual processing priorities is obtained when face processing is assessed in the presence of scrambles as a matched source of noise (Experiment 1) and in a single-stimulus detection task that does not include scrambles (Experiment 3).**

As in Experiment 1, we also examined expression identification. Notably, in Experiment 1 we observed above-chance identification only at exposure durations of above 3 ms – around three times longer than the longest exposure used in the present experiment. Expression identification hinges on the ability to integrate facial features and their configuration; we anticipated that this may require a certain minimal exposure duration regardless of the presence or absence of other stimuli. **Indeed, at this experiment's brief exposures, identification sensitivity was at chance for all durations (Fig. 2B).** No effect or interaction reached significance. (We address minimal exposures for single-stimulus expression identification in Experiment 4, below).

Experiment 3 is described in the revised manuscript as follows (lines 200-230):

“Next, we asked whether waiving this requirement would lead to the same ordering of processing priorities: Would a face's orientation and expression modulate the ability to detect it, even when it is presented on its own? Detecting a single stimulus can be performed using low-level stimulus attributes (without extracting meaning), reducing the exposure required for localisation; therefore, in Experiment 3 we took advantage of our tachistoscope's ability to display stimuli for sub-millisecond durations and selected seven new equally-spaced durations (range 0.25–1.25 ms) covering floor to ceiling localisation performance (see **Supplementary Fig. 10**). The experiment, performed by 32 new

participants, had a very similar design to Experiment 1, but with no scrambled faces. As in Experiment 1, location sensitivity increased with duration (**Fig. 2A**; $F_{(6, 186)} = 300.8, p < .001$), departing from chance level at 0.417 ms for all categories (all $t > 1.93, p < 0.032$). Moreover, even without a matched source of low-level noise (scramble), location sensitivity showed a similar pattern to Experiment 1: A main effect of orientation indicated an FIE, with higher location sensitivity for upright than inverted faces ($F_{(1, 31)} = 4.69, p = .038$); this effect was numerically greater at longer exposure durations, although the interaction between duration and orientation was not significant ($F_{(4.61, 142.9)} = 2.01, p = .087$), suggesting that participants' ability to use low-level attributes for localisation may have diluted the sensitivity-enhancement engendered by engaging face-specific mechanisms; finally, as in Experiment 1, there was no effect of facial expression ($F_{(1, 31)} = 2.03, p = .164$), nor interaction between expression and duration ($F_{(6, 186)} = 1.21, p = .303$), indicating no advantage for emotional compared to neutral faces (see **Supplementary Note 7** and **Supplementary Fig. 11** for more details of Experiment 3).

Fig. 2. Psychophysical measures of location detection and emotion identification in single-stimulus processing. Experiment 3: (A) Location sensitivity departed from chance-level at 0.417 ms. A small but significant upright-face advantage (FIE) was found; there was no effect of facial expression. (B) Emotion identification sensitivity did not depart from chance level at any exposure. **Experiment 4:** (C) Location sensitivity was above chance at all durations, reaching ceiling from 1.7 ms of exposure. (D) Emotion identification sensitivity closely replicated the results of Experiment 1, with a FIE from 5.3 ms of exposure. Error bars: ± 1 SEM.

Unlike Experiment 1, however, these displays were too brief for participants to extract the meaning of facial expressions: identification sensitivity never rose above chance (all $t < 0.781, p > .22$), and no difference was observed between durations (**Fig. 2B**; $F_{(1, 31)} = 1.324, p = .259$). No other effect or interaction reached significance.”

In summary, the results of Experiments 1 and 3 demonstrate **a consistent pattern for face location sensitivity with and without scrambles**. Although the specific minimal durations differ between the two experiments' paradigms, the sequence of required exposures is similar, strongly supporting the generalisability of our findings. Importantly,

single-stimulus detection occurred with much lower exposure durations than in our original Experiment 1; this aligns with standard notions in psychophysics and perception science: stimulus detection involves distinguishing a signal from noise. While the source of noise in our Experiment 1 was a scrambled stimulus that preserved the low-level information of the target face stimulus (requiring extraction of meaning to perform the location task), the source of noise in Experiment 3 was the screen's background colour, which does not match the target stimulus in low-level information. Although the presence of an FIE suggests that location sensitivity was aided by engaging face-specific mechanisms, the overall pattern suggests that participants were able to perform the single-stimulus detection task without necessarily needing to extract meaning from the stimulus. We believe it is important to highlight how these two experiments, in combination, have provided us with a very detailed picture of the processing priority sequence for face detection. In the next experiment, we augment this with a similar clarification for single-stimulus expression identification.

Experiment 4

Since the exposure durations employed in Experiment 3 were too short for expression identification to depart from chance, we ran Experiment 4 to **further explore single-stimulus identification**. While detecting the presence of a stimulus in one of two locations sets the stimulus as the signal and the locations with no stimulus as sources of noise, expression identification works differently: observers must integrate the stimulus features and decide, based on what they know, whether they belong to one category or another. Therefore, we anticipated that expression identification should be unaffected by the presence or absence of low-level-matched scrambles. To test this, Experiment 4 retained the original exposure durations employed in Experiment 1: 0.8, 1.7, 2.6, 3.5, 4.4, 5.3, and 6.2 ms. Experiment 4 was thus a precise replication of Experiment 1 except for the absence of scrambled stimuli.

Consistent with Experiment 3, location sensitivity in Experiment 4 surpassed chance performance across all exposure durations, exhibiting a ceiling effect from 1.7 ms of exposure (**Fig. 2C**). The difference between (above-chance) location sensitivity at 0.8 ms vs all the other durations resulted in a statistically significant effect of exposure duration. No other main effects or interactions reached significance, presumably because of ceiling effects.

Crucially, expression identification **was very similar in this new experiment and in our original Experiment 1; expression identification thus seems unaffected by the absence of scrambled stimuli**. Main effects of exposure duration (greater sensitivity as duration increased) and orientation (greater sensitivity for upright over inverted faces) were modulated by a significant interaction, with the advantage for upright faces (an identification FIE) emerging from 5.3 ms of exposure (new **Fig. 2D**), as in Experiment 1.

Experiment 4 is described in the revised manuscript as follows (lines 235-259):

“Therefore, we ran Experiment 4 in order to establish the minimal durations required for expression identification in single-stimulus displays. Thirty-two new participants were shown the same stimuli as in Experiment 3 (a single face on each trial, upright or inverted and neutral or fearful), but we used the same seven display durations as in Experiment 1 (0.8–6.2 ms). Under these conditions, unsurprisingly, location sensitivity was above

chance for all durations, and at ceiling for all but the shortest duration (**Fig. 2C**), with no effects of orientation or expression and no interactions. For expression identification, we anticipated that performance should be unaffected by the presence or absence of low-level-matched scrambles because this task requires observers to integrate stimulus features and decide which category they belong to. Indeed, observers' ability to identify the expressions in each face closely replicated that seen in Experiment 1 (**Fig. 2D**). Emotion identification increased with duration ($F_{(3.89, 120.665)} = 17.589, p < .001$), with an advantage for upright over inverted faces ($F_{(1, 31)} = 15.99, p < .001$) that interacted with exposure duration ($F_{(6, 186)} = 4.12, p < .001$), arising at exposures of 5.3 ms and above ($t(31) = -5.15, p = .008$), (see **Supplementary Note 8** and **Supplementary Fig. 12** for more details of Experiment 4). Indeed, emotion identification did not differ between Experiment 1 and 4, indicating that the presence or absence of scrambled stimuli did not influence emotion identification (see **Supplementary Fig. 13** for a direct comparison between Experiments 1 and 4). Overall, the sequence of processing priorities established in Experiments 1 and 2 was replicated in Experiments 3 and 4, using single-stimulus, no-scramble displays. Finally, we note that it was not possible to estimate metacognitive sensitivity for these data, because participants provided too few high-visibility ratings at all (Experiment 3) or most (Experiment 4) exposure durations, meaning that there was not enough of the necessary variability in scores for model fitting (see **Supplementary Note 9** and **Supplementary Fig. 14** for details)."

As noted in the text quoted above, we also compared identification performance in Experiments 1 and 4 directly: To test whether the presence or absence of scrambled stimuli had any impact on expression identification, we compared identification sensitivity in Experiments 1 and 4 with a mixed ANOVA, treating the experiment as between-subject factor. As noted in the new **Supplementary Note 8**, similar to the analysis of each experiment on its own, this analysis found main effects of exposure duration and orientation, as well as a significant interaction between these factors. Post-hoc pairwise comparisons confirmed the presence of a FIE from 5.3 ms of exposure in both experiments. Crucially, there was no main effect of experiment, nor any significant interaction with this factor, indicating **no evidence that the presence or absence of scrambles discernibly influenced expression identification sensitivity**. **Supplementary Fig. 13** (reproduced below) shows the identification results of the two experiments superimposed on the same axes and demonstrates their similarity.

This analysis is described in **Supplementary Note 8** as follows:

"To test whether the presence or absence of scrambled stimuli had any impact on expression identification, we compared identification sensitivity in Experiments 1 and 4 with a mixed ANOVA, treating the experiment as between-subject factor, and exposure duration and face orientation as within-subject factors. We found a main effect of exposure duration ($F_{(4.18, 259)} = 30.12, p < .001, \eta^2 = .327$), and a main effect of face orientation ($F_{(1, 62)} = 33.94, p < .001, \eta^2 = .354$). We also found a significant interaction between exposure duration and face orientation ($F_{(6, 372)} = 6.61, p < .001, \eta^2 = .096$). Crucially, we did not find an effect of Experiment ($F_{(1, 62)} = 0.012, p = .912, \eta^2 = 0$). No other interaction reached significance (all $p > .497$). Post hoc pairwise comparisons revealed a significant advantage of upright faces over inverted faces at 5.3 ($t(62) = 4.83, p < .001$) and 6.2 ms of exposure ($t(62) = 6.87, p < .001$). See **Supplementary Fig. 13** for the identification sensitivity results of both experiments superimposed on the same axes. These results indicate that the presence or absence of scrambles did not influence expression identification sensitivity."

Supplementary Fig. 13. Results from Experiment 1 and Experiment 4's identification sensitivity. Experiment 1's expression identification results: Expression identification increased with exposure duration, and an advantage for upright faces was found from 5.3 ms of exposure. Experiment 4's expression identification results: The same pattern of results as in Experiment 1 was found. We found no differences between Experiment 1 and Experiment 4, which indicates that the presence of scrambles did not play a role in expression identification. Error bars: ± 1 SEM. Asterisks denote significant upright-inverted comparisons for Experiment 1 in red, and for Experiment 4 in blue.

We believe that these two new single-stimulus experiments have importantly buttressed and enhanced our understanding of how visual detection and identification of face images unfold. They highlight that the human visual system can detect the location of low-level information presented for less than half a millisecond. They clarify the stimulus priorities that influence this task. And they also show, importantly, that our original findings about the sequence of processing priorities generalise across scramble-present and scramble-absent paradigms, indicating that our use of scrambles did not notably alter the visual processes that participants were using.

Further, we note that even though single-stimulus detection arises with shorter exposure durations (because of the absence of low-level-matched noise), our effects of interest (e.g., impact of orientation on detection and identification) did actually arise in the same order. Finally, our new results show that the presence of scrambles is irrelevant for expression identification, ruling out the possibility that scrambled stimuli might have behaved as distractors for this task. Thus, the presence or absence of scrambles did not qualitatively change what we would conclude from these studies.

We hope that the Reviewer and Editor agree that these new studies, while making an important contribution to the manuscript, also corroborate the usefulness of certain aspects of our original design: Specifically, while it is both interesting and useful to know the temporal limits under which single-stimulus detection or identification can occur, our new experiments' use of extremely brief stimuli without matched noise makes certain aspects of inference and model-fitting harder. For example, these new data could not be fit with model-based estimates of metacognitive sensitivity (which require a broad spread of ratings), because participants provided too few high-visibility ratings at all exposure durations (Experiment 3) or at most of the durations (Experiment 4), meaning that there was not enough of the necessary variability in scores for model fitting (see **Lines 255-259** of the revised manuscript, quoted above; and **Supplementary Note 9**,

reproduced below). Critically, the goal of our EEG studies was to search for neural evidence of unconscious processing (indicated by stimulus-specific activity at shorter exposures than those required for above-zero meta-d'). Further investigations of this neural aim are therefore best served by our use of durations at which meta-d' and behavioural measures of emotion processing (which is unaffected by the presence of scrambles) could be observed.

Information about PAS ratings is described in **Supplementary Note 9**:

“The PAS ratings obtained in Experiments 3 and 4 precluded calculation of metacognitive sensitivity. Calculating meta-d' requires a diverse range of PAS ratings for maximum-likelihood estimate model-fitting, with a sufficiently large number of correct and incorrect trials for each rating^{8,40}. Participants provided too few high (almost clear/clear experience) ratings for all exposure durations in Experiment 3 (**Supplementary Fig. 14A**), and for most durations in Experiment 4 (**Supplementary Fig. 14B**). PAS rating distributions thus lacked an adequate spread for fitting the model-based estimates of metacognitive sensitivity. Only 6.3% (Experiment 3) and 21.9% (Experiment 4) of the participants' data could be fitted for all conditions. This contrasts with the PAS ratings obtained in Experiments 1 and 2, wherein all participants' data could be fitted to the MLE meta-d' model for all conditions.

Supplementary Fig. 14. Mean PAS rating frequencies for Experiments 3 and 4. (A) In Experiment 3, most exposure durations yielded almost no ‘almost clear experience’ and ‘clear experience’ ratings. (B) In Experiment 4, most exposure durations resulted in very low ‘almost clear experience’ and ‘clear experience’ ratings across several durations. Percentages represent overall numbers per exposure durations.

Having empirically addressed the reviewer’s concerns and expanded our understanding of single-stimulus detection using extremely brief exposures, we trust that the reviewer will appreciate these new findings and the highly convergent results across our six studies.

[REDACTED]

Response:

We express our gratitude to the reviewer for their valuable comments. Through careful consideration and incorporation of their feedback, we have enhanced the clarity of key concepts. As a result, our findings now exhibit a heightened clarity, providing a more comprehensive understanding for their interpretation.

[REDACTED]

Response:

We thank the reviewer for their insightful comments and recommendations.

REVIEWERS' COMMENTS

Reviewer #1 (Remarks to the Author):

As a disclaimer, I was not a reviewer on the earlier versions of this manuscript, but was specifically asked by the Editor to gauge to what extent the revised manuscript addresses the issues raised by R1. Although I cannot speak for reviewer 1, in my view, the added single-stimulus experiments successfully address the concerns brought up by the Reviewer.

On a personal note, I find this manuscript to be a very important piece of work, that is likely to give rise to multiple follow-ups in the field. It is very well written, and meticulously designed and analyzed. Also, the conclusions are well-supported by the data. This will be an unusually short review for me.

Very minor points: I would not make this a deal-breaker, but I would like to suggest some minor modifications to the behavioral results graphs, that might make it more readily / intuitively understandable for the reader. (1) add significance (e.g., as a line per condition at the bottom of each graph) indicating deviations from zero sensitivity; (2) add an icon or label indicating what aspect of the stimulus "identification" refers to (in "identification sensitivity"); (3) add a line at 0 sensitivity.

Response to reviewer comments

Reviewer 1

Comment:

As a disclaimer, I was not a reviewer on the earlier versions of this manuscript, but was specifically asked by the Editor to gauge to what extent the revised manuscript addresses the issues raised by R1. Although I cannot speak for reviewer 1, in my view, the added single-stimulus experiments successfully address the concerns brought up by the Reviewer.

On a personal note, I find this manuscript to be a very important piece of work, that is likely to give rise to multiple follow-ups in the field. It is very well written, and meticulously designed and analyzed. Also, the conclusions are well-supported by the data. This will be an unusually short review for me.

Very minor points: I would not make this a deal-breaker, but I would like to suggest some minor modifications to the behavioral results graphs, that might make it more readily / intuitively understandable for the reader. (1) add significance (e.g., as a line per condition at the bottom of each graph) indicating deviations from zero sensitivity; (2) add an icon or label indicating what aspect of the stimulus "identification" refers to (in "identification sensitivity"); (3) add a line at 0 sensitivity.

Response:

We are grateful for the reviewer's positive comments and constructive suggestions. We have implemented the changes suggested to the figures in our main text, by adding: (1) Lines representing statistical significance at the bottom of graphs in Figures 1 and 2; (2) A more accurate y-axis label for identification in Figures 1C, 1F, 2B, and 2D, ("Emotion Identification sensitivity"); and (3) A reference line at zero indicating chance sensitivity in the graphs of Figures 1 and 2.

[REDACTED]